# Using AI Segmentation Models to Improve Foreign Body Detection and Triage from Ultrasound Images

**DOI:** 10.3390/bioengineering11020128

**Published:** 2024-01-29

**Authors:** Lawrence Holland, Sofia I. Hernandez Torres, Eric J. Snider

**Affiliations:** Organ Support and Automation Technologies Group, U.S. Army Institute of Surgical Research, JBSA Fort Sam Houston, San Antonio, TX 78234, USA; lawrence.a.holland7.ctr@health.mil (L.H.); sofia.i.hernandeztorres.ctr@health.mil (S.I.H.T.)

**Keywords:** artificial intelligence, segmentation, triage, ultrasound imaging, shrapnel, foreign body, emergency medicine

## Abstract

Medical imaging can be a critical tool for triaging casualties in trauma situations. In remote or military medicine scenarios, triage is essential for identifying how to use limited resources or prioritize evacuation for the most serious cases. Ultrasound imaging, while portable and often available near the point of injury, can only be used for triage if images are properly acquired, interpreted, and objectively triage scored. Here, we detail how AI segmentation models can be used for improving image interpretation and objective triage evaluation for a medical application focused on foreign bodies embedded in tissues at variable distances from critical neurovascular features. Ultrasound images previously collected in a tissue phantom with or without neurovascular features were labeled with ground truth masks. These image sets were used to train two different segmentation AI frameworks: YOLOv7 and U-Net segmentation models. Overall, both approaches were successful in identifying shrapnel in the image set, with U-Net outperforming YOLOv7 for single-class segmentation. Both segmentation models were also evaluated with a more complex image set containing shrapnel, artery, vein, and nerve features. YOLOv7 obtained higher precision scores across multiple classes whereas U-Net achieved higher recall scores. Using each AI model, a triage distance metric was adapted to measure the proximity of shrapnel to the nearest neurovascular feature, with U-Net more closely mirroring the triage distances measured from ground truth labels. Overall, the segmentation AI models were successful in detecting shrapnel in ultrasound images and could allow for improved injury triage in emergency medicine scenarios.

## 1. Introduction

It is critical in emergency medicine situations to adequately triage casualties to determine how to prioritize limited resources. This is amplified in remote or military medicine, where resources and medical evacuation resources are scarce. Future battlefields will likely involve more contested airspace, further limiting the opportunity for casualty evacuation [1,2]. Medical imaging-based triage is often limited to ultrasound imaging in remote austere environments due to its small size and portability compared to other medical imaging approaches [3]. Ultrasound imaging can allow for the detection of injuries or illnesses such as shrapnel or foreign bodies embedded in tissue [4], COVID-19 lung pathologies [5], pneumothorax, hemothorax, or abdominal hemorrhage [6], which all may be critical to detect during casualty triage.

There are, however, a few challenges with ultrasound-based triage. First, proper image acquisition requires extensive training for the proper positioning of the ultrasound transducer so that a suitable image is acquired. Multiple approaches have been taken to improve ultrasound image acquisition, which include improved standards and protocols, augmented reality overlays while scanning [7], and using robotics to automate image acquisition [8]. Second, image interpretation requires a skilled radiographer to interpret the ultrasound image and make a diagnosis. This type of professional may often not be available in remote or military environments. Third, even with proper image interpretation, there is still a need for objective metrics on which to base triage decisions. Oftentimes these decisions are subjective, but they may be less consistent if taken by less skilled operators in remote emergency scenarios.

Artificial intelligence (AI) models for image interpretation and the subsequent analysis of these models can serve as a basis for simplifying image interpretation and developing objective triage metrics. Deep learning approaches have been extensively applied to ultrasound imaging to automate the detection of various injuries and abnormalities. However, deep learning image interpretation techniques can take different forms. Simple image classification approaches provide a categorical description of the ultrasound image; however, information about regions of interest in the image is not provided, which may be critical for adding granularity to the triage process (Figure 1A). Object detection models provide a higher level of explainability to model predictions by generating bounding boxes around the objects of interest present in an image, such as shrapnel (Figure 1B). We were recently able to use various object detection architectures to identify shrapnel and neurovascular features and use the predictions to measure distances between objects as a preliminary triage metric [9,10]. However, bounding boxes are rectangular shapes that may not properly delineate features at certain angles and may overlook irregular shapes, lessening the accuracy of making triaging predictions from bounding boxes. Instead, different deep learning architectures have been developed for creating segmentation masks for objects of any shape or size (Figure 1C). This allows for higher pixel-wise accuracy when recognizing objects from which triage decisions can be made.

In this work, we first evaluate the feasibility of developing deep learning segmentation models for identifying shrapnel in tissue phantoms as well as performing multi-class segmentation of shrapnel and neurovascular features. Shrapnel or foreign body detection in tissues is especially relevant for improved triages, as data from the Soviet–Afghan war highlighted how 71% of casualties were able to be treated minimally, non-operatively minimizing the need for medical evacuations when these casualties could be properly classified at or near the point of injury [11]. Using the multi-class segmentation model, we develop and improve on a foreign body distance triage metric originally used with object detection AI models.

To accomplish these goals, this paper first reviews how segmentation models have been used for other medical applications. Next, the methods section details how the images were collected and processed as well as the AI model setup for this application. The results section first describes how single-class segmentation models can be used for shrapnel detection followed by expanding these models for multi-class detection of additional neurovascular features. An application is lastly shown to highlight how a multi-class segmentation model can track the distance of shrapnel to neurovascular features as a potential triage tool. 

### Overview of Ultrasound Imaging Artificial Intelligent Segmentation Models

Various edge detection or intensity thresholding methods have been used for medical image segmentation [12]. However, these methods are not fully automated or as robust as more recent AI approaches. Focusing on AI modalities for ultrasound imaging, different architectures have been used, with encoder–decoder neural network approaches being the most widely used, followed by fully convolutional network model architectures [13]. Encoder–decoder segmentation models are similar in structure to more widely used object detection architectures, relying on convolution models to encode the image and extract features, followed by a deconvolution model or decoder which restores the pixel locations for the extracted features.

The most widely used segmentation model for medical imaging is the U-Net encoder–decoder model [14]. U-Net’s name is based on its U-shaped network architecture, which comprises a series of encoder steps to extract features, followed by a series of decoder blocks connected to each encoder by a bridge. The decoder blocks are used to align the extracted features with image pixels and generate segmentation masks. The bridge connections between each encoder and decoder help provide clarity for features identified at each step that may be lost at deeper convolutional steps to improve the accuracy of the segmentation masks. U-Net was developed specifically for biomedical image segmentation, initially for the segmentation of neurons and cell structures in microscope images [14]. It has since been utilized for a wide array of biomedical segmentation tasks including blood vessel, tumor, and organ segmentation [12,13,15].

Another widely used architecture for image segmentation is the YOLO (You Only Look Once) series of architectures [16]. YOLO models are single-stage object detectors made up of a convolutional backbone, where features are extracted, a neck, where features are mixed and combined, and a head, where locations and classes of objects are predicted. One of the latest models in this series is YOLOv7 [17]. YOLOv7 improves upon previous architectures by updating the backbone to include Extended Efficient Layer Aggregation (E-ELAN). This and several other improvements have served to make the model both more accurate and faster than previous YOLO releases. While YOLO is traditionally an object detection framework, YOLOv7 has been modified for instance segmentation use cases [17]. This architecture has been used to develop segmentation models for self-driving cars [18] as well as some medical image segmentation applications [19,20].

Image segmentation AI models can have many benefits for medical image interpretation, aiding with the recognition of regions of interest which could impact treatment and diagnosis [21,22]. As mentioned previously, U-Net was purpose-built for medical imaging. In certain emergent conditions the prompt interpretation of medical imaging can be critical, particularly when highly specialized personnel are not available. For this reason, U-Net architecture has been used to identify pleural effusion [23], measure the optic nerve sheath [24], and diagnose ascites [25]. The common use of U-NET for the segmentation of biomedical imaging has led to architecture modification that better fit specific areas and imaging modalities [26,27]. Along this line, segmentation architectures such as YOLOv7 have been used to achieve a reliable performance for the diagnostic segmentation of fetal cardiac features [28], kidney stones [29], and cervical lymph nodes [30].

## 2. Materials and Methods

### 2.1. Phantom Preparation and Imaging

The shrapnel image dataset was collected from an ultrasound tissue phantom that has been extensively used to train multiple AI models [31]. Briefly, a double-layer gelatin phantom was made by dissolving gelatin 10.0% *w*/*v* (Thermo-Fisher, Waltham, MA, USA) and flour (0.25% *w*/*v* inner layer and 0.10% *w*/*v* outer layer, unbleached wheat flour, H-E-B, San Antonio, TX, USA) in an evaporated milk (Costco, Seattle, WA, USA) and water mixture at a 1:2 ratio, respectively. Fragments made of agarose (Fisher Scientific, Fair Lawn, NJ, USA) at 2.0% *w*/*v*, with varying flour concentrations of 0.10% and 1.0% *w*/*v*, were added to the inner layer of the phantom in order to add training noise between different phantoms. A total of six phantom subjects were created to generate the training data: half of the subjects were a simple phantom with no additional features, and the other half were complex phantoms with an additional neurovascular bundle. The neurovascular bundle consisted of three channels created with a biopsy punch, which were left hollow to simulate the vein and artery and filled with the gelatin mixture with flour at 0.5% *w*/*v* for the nerve channel. Shrapnel insertion and imaging of the phantom were performed under water in 10 s clips. A 2.0 mm diameter brass rod (McMaster Carr, Elmhurst, IL, USA) was cut into pieces of five different lengths (1.0 cm, 0.8 cm, 0.6 cm, 0.4 cm, and 0.2 cm), that were used as shrapnel. All the ultrasound images were collected using a Sonosite Edge II System (Fujifilm, Bothell, WA, USA) with the HL50x (Fujifilm, Bothell, WA, USA) transducer.

### 2.2. Preprocessing and Labeling Images

After the ultrasound videos had been recorded, individual frames were extracted as images, totaling 12,144 collected ultrasound images. Next, the images were manually annotated with the locations of neurovascular features and shrapnel, with 6385 receiving annotations only for shrapnel and the remaining 5759 being annotated with vein, artery, nerve, and shrapnel features. From these, 15 and 59 images were split to be used as holdout test images after the models were trained for the single-class and multi-class models, respectively. The annotations were generated manually using the COCO annotator software to label pixels corresponding to the desired features in each image. The annotations were then saved to a JSON file containing file and annotation information for each image.

Before the training of the models could start, Python scripts were developed to read the COCO JSON file and use it to produce images and masks compatible with the needs of the segmentation models. In the case of U-Net, this involved resizing all the images to 256 × 256 and generating image masks from the original COCO data. These masks consist of one-channel 256 × 256 images with each desired class represented by a unique integer value (such as vein = 1, artery = 2, etc.). For YOLOv7 training, the images were kept in their original 512 × 512 size, and the COCO annotation data were used to generate text files for each image that contain segmentation information for each class. 

### 2.3. Training Segmentation Algorithms

All training was conducted using the same computer hardware setup: an NVIDIA GeForce RTX 3090 Ti 24 GB VRAM system with Intel i9-12900k and 128 GB RAM with a Windows 10 Pro operating system. YOLOv7 training was completed using an Anaconda distribution running Python 3.8.8. U-Net training was conducted using MATLAB (R2023a, Mathworks, Natick, MA, USA) with deep learning, computer vision, and image processing packages installed. Before training, the images were randomly shuffled and separated into training and validation sets at an 80:20 ratio and paired with their respective masks.

A publicly available version of the YOLOv7 segmentation model was used [17]. Minor changes were made to the training and prediction scripts of the original code to be able to extract generated prediction masks into text files for each image per YOLOv7 requirements. The ultrasound images were paired with their respective segmentation text file containing ground truth information prior to training. The model was trained for a total of ten epochs, recording precision, recall, loss, and mean average precision for each epoch.

The U-Net model was implemented in MATLAB, which provides a ready-made 4-depth U-Net for computer vision tasks. The customizations for our purposes included removing the final layer of the model and replacing it with a pixel classification layer; batch normalization layers were added after each convolution. Batch normalization works by normalizing the output of each convolutional layer, allowing for a deeper and faster running model and helping to prevent overfitting by inserting a small amount of noise into the data. The original U-Net publication included max pooling layers after each convolution layer. However, these were not present in the MATLAB implementation of U-Net. The U-Net model was trained for a total of twenty epochs, with accuracy and loss values being recorded after processing each batch of images. Similar training approaches were taken with both architectures when training a single-class model for shrapnel and the multi-class models further including neurovascular classes.

### 2.4. Performance Metrics for Evaluating Trained Models

For each of the trained models, prediction segmentation masks were generated from a separate set of blind images. From these, every pixel in the prediction mask was compared to the ground truth masks, and labeled as either True Positive (TP), False Positive (FP), True Negative (TN), or False Negative (FN). The total number of each label was compiled for each class as well as for the overall image and used to calculate performance metrics including accuracy, precision, recall, the intersection-over-union (IOU) score, and the Dice coefficient. Formulas for all the metrics used are shown in Table 1. Accuracy is a measure of how frequently a model can be expected to make correct predictions. Precision measures how often positive predictions are correct, while recall shows the efficiency of a model to accurately predict positive features. The IOU score and Dice coefficient are common metrics in object detection and segmentation models that determine how well the model prediction masks correspond to the ground truth masks.

### 2.5. Distance Triage

The distance between shrapnel and neurovascular bundle anatomical features was calculated as it could be a useful metric for triaging injury severity. This analysis was performed using segmentation masks generated from the predictive models. The perimeter of each object (shrapnel, nerve, artery, and vein) was found after isolating each object.

Distance triage is calculated from the predicted shrapnel to the predicted vein, artery, or nerve, and then the distance from the shrapnel to the closest structure is provided. This was performed using MATLAB’s regionprops functionality to find the closest distance of shrapnel to each neurovascular feature, as well as combinations of multiple classes if multiple shrapnel or nerves were identified. This was achieved by comparing each pixel in the boundary of each anatomical feature to every pixel value in the boundary of the shrapnel segmentation mask to find the two closest points between the two (given as a pair of X,Y coordinates). After these were identified, the minimum of all the options was selected, which served as the triage score for this application (Figure 2). Multi-class segmentation masks (*N* = 46) generated by the AI models were compared to their respective ground truth masks to measure the accuracy of each model for triage. The outlier exclusion ROUT (robust regression followed by outlier identification) methodology was applied to the measured distance error to remove data points with a false discovery rate set at 1%. Statistical analysis and figures were generated using GraphPad Prism 10.1.2 (La Jolla, CA, USA).

## 3. Results

### 3.1. Single-Class Segmentation Models for Shrapnel

For shrapnel segmentation, the YOLOv7 model achieved an 82.2% validation precision in nine training epochs, whereas U-Net reached a validation accuracy of over 90% in the first epoch and continued to increase steadily throughout training, reaching a final validation accuracy of 99%. From the test set for the single-class shrapnel segmentation models, the accuracy, precision, recall, IOU, and Dice coefficient scores were calculated by comparing the predicted shrapnel masks to the ground truth masks (Table 2). Both models performed alike in identifying shrapnel in the test images, with both receiving an accuracy of 99.1%. Both models also received similar scores in all the other recorded metrics, with YOLOv7 slightly outscoring or equaling U-Net in all the metrics. Representative results of both models on the same ultrasound image are shown in Figure 3.

### 3.2. Multi-Class Segmentation Models

The YOLO model was trained for ten epochs, reaching a best precision of 84.8% and recall of 84.2%. The U-net model reached an accuracy value over 90% in the fourth epoch, steadily increasing after reaching a final accuracy of 96.5% at the end of twenty training epochs. By visual comparison of the predicted masks for both models to the ground truth masks, the U-Net model produced generally larger segmentation masks for each feature than the YOLOv7 model, with the size of the ground truth masks falling in-between the two (Figure 4). In some instances, the U-Net model would produce segmentation masks with two or more class prediction masks in contact with one another or even overlapping.

To assess the performance of both models, metrics were calculated for each prediction class and image background classification, with artery features having the highest test accuracy at more than 99.0% for each model (Table 3). Averaging across all the features, both models achieved accuracy scores above 98.0% for all classes (Figure 5). However, recall was much higher for U-Net than YOLOv7 at 92.6% vs. 82.7%, in contrast to precision, which was higher for YOLOv7 than U-Net, at 77.7% and 66.2%, respectively. 

In addition, we evaluated performance excluding the image background class as it was the most prevalent feature in all the images. Focusing on only the features, YOLOv7 outscored U-Net for test precision, with a difference of 14.6% between the two models. Conversely, U-Net outperformed YOLOv7 with regard to test recall, with a difference of 12.8%. Both models received similar scores for the IOU and Dice coefficient, with YOLOv7 slightly outscoring U-Net for the IOU by 4.3% and the Dice coefficient by 2.5%. 

In contrast to the single-class shrapnel model, the U-Net multi-class model precision fell by a value of 0.201 while recall increased by 16.4% for the shrapnel predictions. The IOU and Dice coefficient metrics decreased slightly between shrapnel-only and multi-class U-Net by 8.0% and 6.5%, respectively. YOLOv7 saw similar trends, with precision falling by 18.5% and recall increasing by 4.5%. The accuracy and IOU saw almost no change, while the Dice coefficient fell by 11.1% for the YOLOv7 multi-class model.

### 3.3. Triage Metric for Segmentation

For the triage metric, minimum distances between each feature and shrapnel were calculated for all the test images, using both the ground truth and prediction masks (Figure 6). For the ground truth labels, the average minimum distance between the shrapnel and the closest anatomical feature was found to be 10.2 pixels, with a standard deviation of 10.8 pixels (Figure 7). The average minimum distances for both U-Net and YOLOv7 were 3.8 and 9.7 pixels, respectively. For each predicted image, the minimum distance triage metric was compared to its corresponding ground truth label, with the difference between the two distances (difference) and the percentage error (% error) being calculated for each image. These values were then averaged across all the test images to obtain a broad view of the distance triage performance. The average difference for the minimum distances between the U-Net predictions and the ground truth for all the images was 7.7 pixels and an average error of 72.5%. For the YOLOv7 predictions, a 5.1-pixel difference was calculated at an average error of 45.5% (Figure 7).

## 4. Discussion

Given the small form factor, low power requirements, and portability of ultrasound technology, it can be a valuable tool for providing more objective criteria for medical triage at or near the point of injury. This is especially critical in remote medical scenarios where resources are limited and medical evacuation can only be made available to the most severe injuries, such as during combat casualty care. However, ultrasound imaging is only valuable if the images can be easily and objectively interpreted for making medical triage decisions. The artificial intelligent segmentation models proposed here for identifying shrapnel and neurovascular features highlight how AI technology can help standardize medical decision making.

For this effort, we evaluated the use of two segmentation models—YOLOv7 segmentation and U-Net—for tracking shrapnel only or multi-class applications including vein, artery, and nerve features. Both models were successful in identifying shrapnel and neurovascular features using the thousands of training images collected in a custom-made ultrasound tissue phantom. However, there were performance differences between the two models, notably with precision and recall. Precision, a measurement of the accuracy of positive predictions, was significantly higher with YOLOv7, while recall, a measurement of accurately identifying positive features, was higher with U-Net. This was mostly due to the prediction masks around the identified features in the YOLOv7 model were fitted around the object perimeter, resulting in more false-negative pixels but few false-positive pixels, while U-Net generally had larger prediction masks resulting in an opposite false bias. Which bias is preferred for medical triage tasks such as this one is unknown. The smaller YOLOv7 prediction masks would improve accuracy in identifying the exact shape of the shrapnel object, which could be critically important for aiding surgical removal, while U-Net would provide a more cautious triage outlook by overestimating the closeness of the shrapnel to neurovascular features (triage distances of 3.8 and 9.7 pixels for U-Net and YOLOv7, respectively). However, either segmentation model likely provides a much higher fidelity for the determination of distances between shrapnel and neurovascular features than the object detection models we previously used for this application. Bounding boxes may be inferior for labeling the irregularly shaped geometries that can often be expected with foreign-body objects embedded in tissues.

It is worth noting that the determination of model performance for both AI models is highly dependent on the training dataset. While attempts were made to add subject variability to the phantom creation process, the tissue phantom used to collect the shrapnel image dataset lacks tissue-level organization, as seen in animal tissue. In addition, the ground truth labels are biased by the image labeler, which can be prone to error. Given the quantity of images being labeled (over 20k images), only a single label was drawn for each US image. For a more robust training set, multiple labelers should be pooled to remove labeling subjectivity from the process. This is especially critical in the case of YOLOv7’s performance, which oftentimes focused on accurately defining the edge of labeled features compared to the ground truth mask. Additional US images will be needed to expand on the training of these segmentation models prior to real-time deployment for use with animal or human triage processing. The translation of this effort will require engagement with the FDA to ensure that the regulatory concerns associated with AI technologies are properly addressed.

## 5. Conclusions

In summary, medical imaging-based triage is a critical tool for accurate decision making regarding injury severity and evacuation priorities if the images can be properly interpreted. Image interpretation challenges are compounded in remote emergency medical situations, such as combat casualty care, due to limited medical personnel and resource availability. The segmentation AI models developed in this work were successful in identifying both foreign-body objects and neurovascular features. This allowed for the measurement of the proximity of the shrapnel to artery, vein, or nerve features as a means of objectifying the triage criteria for this application. Further work is needed to train the AI models using *in-vivo* medical images rather than *in-vitro*, and the triage framework can be expanded to other applications of relevance to emergency medicine. This will improve the prioritization of limited resources at or near the point of injury and improve evacuation priorities during remote emergency medical situations.

## Figures and Tables

**Figure 1 bioengineering-11-00128-f001:**
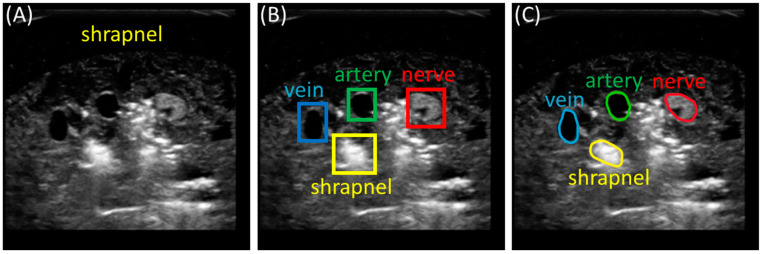
Comparison of AI model types for shrapnel and neurovascular feature identification. Representative ultrasound image showing output differences between a (**A**) binary classification model, (**B**) a multi-class object detection model, and (**C**) a multi-class segmentation model. The classification model labels images as either positive or negative for shrapnel. The object detection model makes predictions and places a bounding box around the predicted objects. The segmentation model creates an outline around the detected object.

**Figure 2 bioengineering-11-00128-f002:**
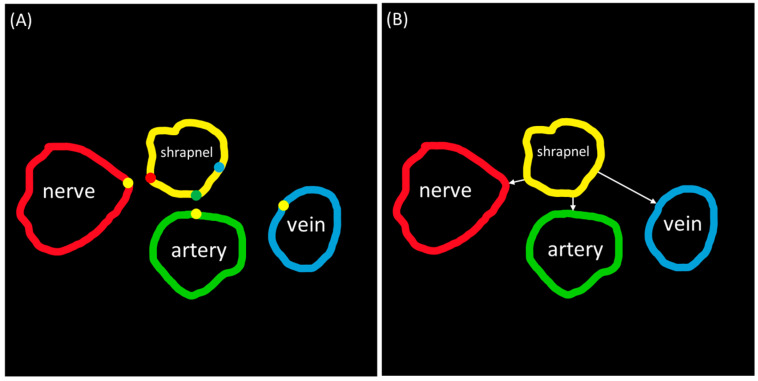
Diagram of triage metric calculation. (**A**) Each pixel from the boundary of each vein (blue), artery (green), and nerve (red) are compared to boundary pixels of the shrapnel (yellow) to find the closest two points (given as a pair of X,Y coordinates), with the closest distance pairs represented by the different dots. (**B**) The distance between the closest point on each feature and the shrapnel are calculated using the distance formula, shown as white arrows from shrapnel mask.

**Figure 3 bioengineering-11-00128-f003:**
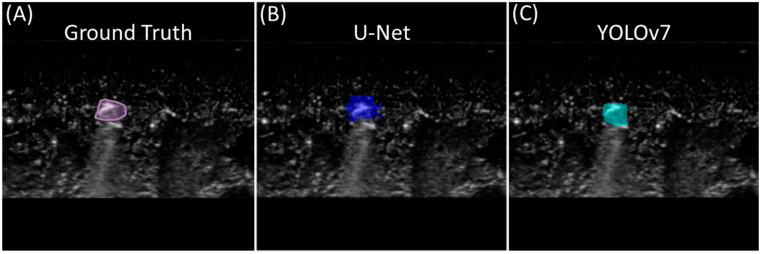
Single-class segmentation for shrapnel. Representative image showing (**A**) ground truth mask, (**B**) U-Net-model-generated mask, and (**C**) YOLOv7-model-generated mask for the same shrapnel image.

**Figure 4 bioengineering-11-00128-f004:**
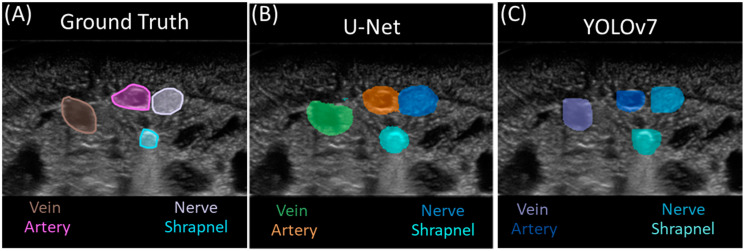
Multi-class segmentation for shrapnel. Representative image showing (**A**) ground truth mask, (**B**) U-Net-model-generated mask, and (**C**) YOLOv7-model-generated mask for the same ultrasound image. All classes are identified in each image.

**Figure 5 bioengineering-11-00128-f005:**
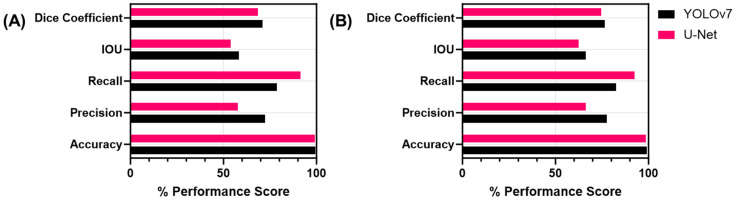
Summary of performance metrics for multi-class segmentation. (**A**) Average metrics for vein, artery, nerve, and shrapnel features and (**B**) performance metrics including the background.

**Figure 6 bioengineering-11-00128-f006:**
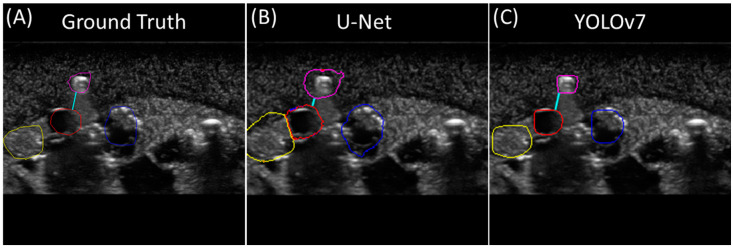
Distance triage metric results. Representative image showing (**A**) ground truth mask, (**B**) U-Net-model-generated mask, and (**C**) YOLOv7-model-generated mask for the same ultrasound image. The light blue line shown indicates the generated minimum distance between the shrapnel (shown in pink) and the closest anatomical feature: nerve (yellow), artery (red), or vein (dark blue).

**Figure 7 bioengineering-11-00128-f007:**
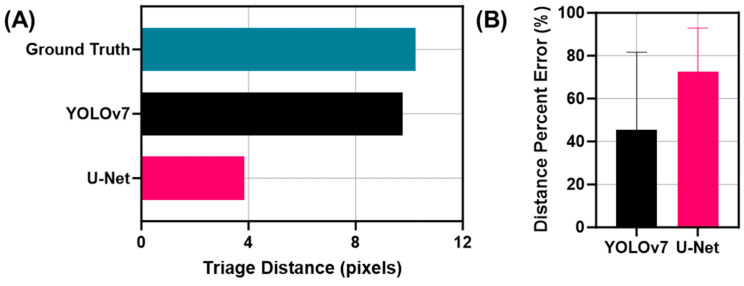
Summary of triage metrics for multi-class segmentation. (**A**) Average minimum triage distance measurements for ground truth, YOLOv7, and U-Net (*N* = 46 images). (**B**) Percent error calculations relative to the ground truth distances for YOLOv7 (*N* = 41 after outliers removed) and U-Net (*N* = 38 after outliers removed).

**Table 1 bioengineering-11-00128-t001:** Formulas for performance metrics. Each performance metric is calculated using True Positives (TP), False Positives (FN), True Negatives (TN), and False Negatives (FN) predictions.

Metric	Formula
**Accuracy**	(TP + TN)/(TP + FN + TN + FP)
**Precision**	TP/(FP + TP)
**Recall**	TP/(FN + TP)
**IOU**	TP/(TP + FP + FN)
**Dice Coefficient**	2 × TP/(2 × TP + FP + FN)

**Table 2 bioengineering-11-00128-t002:** Summary of performance metrics for single-class shrapnel segmentation.

	Accuracy	Precision	Recall	IOU	Dice Coefficient
**YOLOv7**	99.1%	76.0%	74.3%	55.1%	69.7%
**U-Net**	99.1%	63.8%	67.7%	51.5%	64.7%

**Table 3 bioengineering-11-00128-t003:** Summary of performance metrics by class for the segmentation models.

	**YOLOv7**				
	**Accuracy**	**Precision**	**Recall**	**IOU**	**Dice Coefficient**
**Vein**	99.4%	75.9%	76.2%	61.6%	74.0%
**Artery**	99.5%	79.2%	78.3%	64.0%	76.8%
**Nerve**	99.4%	76.9%	81.4%	61.9%	74.9%
**Shrapnel**	99.2%	57.5%	78.8%	45.7%	58.6%
**Background**	97.8%	99.0%	98.7%	97.7%	98.8%
	**U-Net**				
	**Accuracy**	**Precision**	**Recall**	**IOU**	**Dice Coefficient**
**Vein**	99.2%	60.5%	85.2%	54.7%	69.9%
**Artery**	99.4%	72.1%	84.1%	63.7%	77.1%
**Nerve**	98.9%	55.2%	97.4%	54.3%	69.5%
**Shrapnel**	98.9%	43.7%	84.1%	43.5%	58.2%
**Background**	97.1%	99.8%	97.1%	96.9%	98.5%

## Data Availability

The datasets generated during and/or analyzed during the current study are available from the corresponding author upon reasonable request.

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
