# Peer review of "Using AI Segmentation Models to Improve Foreign Body Detection and Triage from Ultrasound Images"

_bioengineering, 2024, doi:10.3390/bioengineering11020128_

Round 1

Reviewer 1 Report

Comments and Suggestions for Authors

The manuscript addresses the problem of identifying shrapnel in tissue as well as multi-class segmentation and detection of shrapnel and neurovascular features such as vein, artery, or nerve using two state-of-the-art deep learning models, YOLOv7 and U-Net. The efficiency of the models was evaluated using accuracy, precision, recall, IOU, and dice coefficient metrics. Additionally, the triage metric for segmentation is introduced and presented.

1. Please add a paragraph highlighting the detailed structure of the paper at the end of the introduction chapter.

2. The last paragraph of subsection 2.3 presents the customization that were applied to the U-Net architecture, such as replacing the last layer with a pixel classification layer or using batch normalization layers. The authors may consider creating a figure that highlights the changes that were made to the model as well as explaining the benefits of using the batch normalization layer after each convolutional layer.

3. Please express the values reported in Table 2–Table 4 in percentages (e.g., 0.994 to 99.4%) to enhance readability. Additionally, specify in the header of Table 5 that the unit of measure for distance and standard deviation is pixels.

4. In the last paragraph of subsection 3.2 (lines 238-245), the resulted performance metrics are expressed in percentages, while in the second, third, and fourth paragraphs, they are written as decimal numbers. Please be consistent with measurement units. For enhanced readability, you may express the values in percentages.

Author Response

The manuscript addresses the problem of identifying shrapnel in tissue as well as multi-class segmentation and detection of shrapnel and neurovascular features such as vein, artery, or nerve using two state-of-the-art deep learning models, YOLOv7 and U-Net. The efficiency of the models was evaluated using accuracy, precision, recall, IOU, and dice coefficient metrics. Additionally, the triage metric for segmentation is introduced and presented.

  1. Please add a paragraph highlighting the detailed structure of the paper at the end of the introduction chapter.

Thanks for the thorough review of our manuscript. We have added this suggested edit into the introduction section to better explain how the rest of the paper will be structured. This begins on line 84.

  1. The last paragraph of subsection 2.3 presents the customization that were applied to the U-Net architecture, such as replacing the last layer with a pixel classification layer or using batch normalization layers. The authors may consider creating a figure that highlights the changes that were made to the model as well as explaining the benefits of using the batch normalization layer after each convolutional layer.

Thank you for your feedback. An explanation for the use of batch normalization was added, starting at line 191.

  1. Please express the values reported in Table 2–Table 4 in percentages (e.g., 0.994 to 99.4%) to enhance readability. Additionally, specify in the header of Table 5 that the unit of measure for distance and standard deviation is pixels.
  2. In the last paragraph of subsection 3.2 (lines 238-245), the resulted performance metrics are expressed in percentages, while in the second, third, and fourth paragraphs, they are written as decimal numbers. Please be consistent with measurement units. For enhanced readability, you may express the values in percentages.

All results expressed as ratios have been changed to percentages when possible in both the results tables and the written material.

Reviewer 2 Report

Comments and Suggestions for Authors

Ultrasound technique is not intended for interpretation of foreign body or other changes in the lungs, because it is difficult for ultrasound waves to pass through lung parenchyma due to the air in the lungs. 

Author Response

Ultrasound technique is not intended for interpretation of foreign body or other changes in the lungs, because it is difficult for ultrasound waves to pass through lung parenchyma due to the air in the lungs. 

We appreciate the concern raised by the reviewer and agree that ultrasound would be an ill-advised imaging approach for air filled tissue. However, that is not the proposed application in this manuscript. Here, we were looking at how segmentation AI models can track shrapnel in tissue. The tissue is mostly mimicking thigh tissue with a central femoral artery and vein present in the tissue phantom used. This tissue is not an air-filled tissue. We have used AI models for diagnosing injuries in the pleural space but never deeper into the lung parenchyma. We hope these clarifications clear up the raised concern.

Reviewer 3 Report

Comments and Suggestions for Authors

The work "Using AI Segmentation Models to Improve Foreign Body Detection and Triage from Ultrasound Images" focuses on utilizing YOLOv7 and U-Net to improve emergency medical ultrasound picture interpretation. The article shows how these algorithms can identify shrapnel and other foreign bodies in tissue for military or distant medical triage.

Comments and Improvements:

1. Main Point and Evidence: The paper persuasively argues for AI in medical imaging using quantitative model performance statistics. Real-world talks of YOLOv7 and U-Net's benefits are scarce. More case studies or real-world examples would show how these models benefit emergency medical situations.

2. Overall Structure: The work is well-structured, but comparison analyses or graphs showing model performance disparities would make the results section more impactful. Add visual aids or comparative charts to help readers grasp the results.

3. Style and intellectual Standards: The work is well-written and intellectual. Adding more background about AI's involvement in emergency medicine would improve the introduction. Add a brief description of AI applications' evolution and current state in emergency medical imaging to the introduction.

4. Cohesion and Coherence: Section transitions, especially technique to outcomes, might be smoother. Suggestion: Use transitional sentences to show how the technique directly affects the results.

5. Addressing Blind Spots: The report should highlight the limitations and potential obstacles of using these AI models in real life. Consider adding a section on regulations and future research to handle model deployment and data privacy issues.

The paper could explore these topics to better understand AI models' practical ramifications and emergency medical care effects.

Author Response

The work "Using AI Segmentation Models to Improve Foreign Body Detection and Triage from Ultrasound Images" focuses on utilizing YOLOv7 and U-Net to improve emergency medical ultrasound picture interpretation. The article shows how these algorithms can identify shrapnel and other foreign bodies in tissue for military or distant medical triage.

Comments and Improvements:

  1. Main Point and Evidence: The paper persuasively argues for AI in medical imaging using quantitative model performance statistics. Real-world talks of YOLOv7 and U-Net's benefits are scarce. More case studies or real-world examples would show how these models benefit emergency medical situations.

Thanks for the suggestion. We have added more use cases, especially emergency medicine focused starting at line 124 for both the YOLO and U-Net architecture used in this effort.

  1. Overall Structure: The work is well-structured, but comparison analyses or graphs showing model performance disparities would make the results section more impactful. Add visual aids or comparative charts to help readers grasp the results.

As recommended, we have converted two of the tabular results into figures. They are at line 303 and 324 in the results section.

  1. Style and intellectual Standards: The work is well-written and intellectual. Adding more background about AI's involvement in emergency medicine would improve the introduction. Add a brief description of AI applications' evolution and current state in emergency medical imaging to the introduction.

See response to point 1

  1. Cohesion and Coherence: Section transitions, especially technique to outcomes, might be smoother. Suggestion: Use transitional sentences to show how the technique directly affects the results.

We have reviewed the methods and results and have added transitional sentences and statements where possible for the content of the manuscript.

  1. Addressing Blind Spots: The report should highlight the limitations and potential obstacles of using these AI models in real life. Consider adding a section on regulations and future research to handle model deployment and data privacy issues.

We agree with the reviewer that these are limitations with any AI medical research application, but not the direct focus of this manuscript. Instead, we have added information on these future challenges in the discussion (Line 387) and add citations to the publication more focusing on these challenges.

The paper could explore these topics to better understand AI models' practical ramifications and emergency medical care effects.

Round 2

Reviewer 2 Report

Comments and Suggestions for Authors

The corrections improved the manuscript.